# Risk Prediction Performance of the Thai Cardiovascular Risk Score for Mild Cognitive Impairment in Adults with Metabolic Risk Factors in Thailand

**DOI:** 10.3390/healthcare10101959

**Published:** 2022-10-07

**Authors:** Nida Buawangpong, Chanchanok Aramrat, Kanokporn Pinyopornpanish, Arintaya Phrommintikul, Atiwat Soontornpun, Wichuda Jiraporncharoen, Suphawita Pliannuom, Chaisiri Angkurawaranon

**Affiliations:** 1Department of Family Medicine, Faculty of Medicine, Chiang Mai University, Chiang Mai 50200, Thailand; 2Global Health and Chronic Conditions Research Group, Chiang Mai University, Chiang Mai 50200, Thailand; 3Division of Cardiology, Department of Internal Medicine, Faculty of Medicine, Chiang Mai University, Chiang Mai 50200, Thailand; 4Division of Neurology, Department of Internal Medicine, Faculty of Medicine, Chiang Mai University, Chiang Mai 50200, Thailand

**Keywords:** mild cognitive impairment, metabolic syndrome, cardiovascular risk, primary care, screening

## Abstract

Individuals with metabolic risks are at high risk of cognitive impairment. We aimed to investigate whether the Thai Cardiovascular Risk (TCVR) score can be used to predict mild cognitive impairment (MCI) in Thai adults with metabolic risks. The study was conducted using secondary data of patients with metabolic risks from Maharaj Nakorn Chiang Mai Hospital. MCI was indicated by an MoCA score of less than 25. Six different TCVR models were used with various combinations of ten different variables for predicting the risk of MCI. The area under the receiver operator characteristic curve (AuROC) and Hosmer–Lemeshow goodness of fit tests were used for determining discriminative performance and model calibration. The sensitivity of the discriminative performance was further evaluated by stratifying by age and gender. From a total of 421 participants, 348 participants had MCI. All six TCVR models showed a similar AuROC, varying between 0.58 and 0.61. The anthropometric-based model showed the best risk prediction performance in the older age group (AuROC 0.69). The laboratory-based model provided the highest discriminative performance for the younger age group (AuROC 0.60). There is potential for the development of an MCI risk model based on values from routine cardiovascular risk assessments among patients with metabolic risks.

## 1. Introduction

Individuals with metabolic disease are not only at high risk of cardiovascular events but also cognitive decline. The significant risk factors are also components of metabolic risk, including high blood pressure, elevated blood sugar, elevated waist circumference (WC), and abnormality of plasma cholesterol or triglyceride levels [1,2]. Many studies have demonstrated that these conditions increase the risk of major cardiovascular diseases via direct damage to vessel walls or systemic changes [3,4,5]. Major cardiovascular events, including myocardial infarction and strokes, are a significant cause of death in this population worldwide.

In addition, a significant number of individuals with metabolic diseases also experience cognitive deterioration [6,7,8]. There is growing evidence to suggest that abnormal metabolic parameters and subsequent cardiovascular diseases are linked to a higher risk of cognitive decline. High blood pressure could cause cognitive deterioration through vascular injury, blood vessel remodeling, and a reduction in cerebral perfusion and may affect brain shape and function [9,10,11]. High plasma glucose increases the risk of cognitive impairment by increasing neuronal insulin resistance, inducing a pro-inflammatory state, impairing mitochondrial function, and causing vascular damage [12]. In addition, hyperlipidemia could cause endothelial damage and aggravate inflammation, resulting in cognitive decline [13]. Elevated WC is related to insulin resistance, which is associated with poor cognitive performance in people with metabolic risks [14,15]. In association with these risk factors, a patient can progress from normal cognitive function to mild cognitive impairment (MCI) and then dementia [16]. MCI is the condition that lies between normal cognitive aging and dementia. It is defined as evidence of progressive cognitive decline combined with the preservation of functional independence [17]. MCI in metabolic syndrome is higher when compared to the general population [18]. The rate of progression depends on the pathogenesis of cognitive decline. The progression of cognitive change in patients with vascular dementia could be faster than in those with Alzheimer’s disease [19].

Although there is a link between the risk factors for major cardiovascular events and cognitive impairment, screening for MCI in people with metabolic diseases is less common than performing cardiovascular risk assessment [20]. Standard tools for the estimation of cardiovascular risk have been developed over recent decades [21,22,23]. In Thailand, healthcare providers are using cardiovascular risk assessment tools for patients with metabolic risk factors, but MCI risk assessment tools are not available for this population, despite evidence that the presence of cardiovascular risk factors could be associated with a high risk of developing cognitive decline. The current screening tool commonly used for the detection of MCI which has a high sensitivity and specificity is the Montreal Cognitive Assessment tool (MoCA) test [24]. However, one of the possible limitations of using this tool is that it is too lengthy to be used as a brief screening tool in normal clinical practice [25]. For this reason, the study aimed to develop predictive models for cognitive impairment derived from variables commonly obtained from routine cardiovascular (CV) risk assessment. This could be useful for the improvement of clinical practice and the identification of people with a high risk of developing MCI.

## 2. Materials and Methods

### 2.1. Dataset and Participants

This study utilized data collected from a previous study examining the possible link between fibroblast growth factor 21 and cognitive decline among patients with metabolic risk factors [26]. Additional data were collected using a similar methodology for cardiovascular risk assessment and cognitive assessment to increase the sample size and power between 1 May 2020 and 31 May 2021. All methods were carried out in accordance with relevant guidelines and regulations. The study protocols were approved by the Research Ethics Committee of the Faculty of Medicine, Chiang Mai University, for the previous study (No. 02671) and additional data collection (No. 08246). The subjects were recruited from the outpatient clinics at Maharaj Nakorn Chiang Mai Hospital which provide care for patients with metabolic risks. These clinics included the family medicine clinic, general medicine clinic, endocrinology clinic, and cardiology clinic. Subjects had not been diagnosed with MCI prior to the enrollment period. We included patients 45 years old or over with at least one metabolic risk factor, including WC over 40 inches for men or 35 inches for women; a body mass index of 25 kg/m2 or over; blood pressure over 130/85 mmHg; fasting triglyceride level over 150 mg/dL; fasting high-density lipoprotein cholesterol (HDL) level less than 40 mg/dL for men or 50 mg/dL for women; and fasting blood sugar over 100 mg/dL [20]. All eligible patients were reviewed for their health assessment which included measurements of systolic blood pressure (SBP), WC, height, MoCA score, and laboratory results, including total cholesterol (TC), HDL, and low-density lipoprotein cholesterol (LDL). Blood tests were performed within the six months prior to the cognitive assessment. Patients with a prior diagnosis of dementia, a positive screening for dementia or depression, or a history of previous brain surgery were excluded. Informed consent was obtained from all subjects.

### 2.2. Cognitive Assessment by MoCA Score [27]

MoCA is the standard tool for screening patients who have MCI. It is a 30-question test. The MoCA is recommended by the Canadian Consensus Conference for Diagnosis and Treatment of Dementia Guidelines for Alzheimer’s disease, and the National Institutes of Health and Canadian Stroke Consortium for Vascular Cognitive Impairment and is available in more than 200 countries around the world [27]. It is used to assess several cognitive domains including attention/concentration, visuospatial/executive functions, memory, language, conceptual thinking, naming, and orientation. The maximum score is 30 and the cut-off score is 24/25 [28]. A score lower than 25 indicates MCI.

### 2.3. Thai Cardiovascular Risk (TCVR) Score Models

The TCVR score was developed to estimate the ten-year risk of a major cardiovascular event in the Thai population based on the data of the patient which included age, gender, diabetic status (DM), SBP, WC, height, TC, HDL, and LDL. The TCVR can be calculated using six models. The first two models do not require any laboratory data and the last four models require some combination of lipid profiles. Six different models were generated to encourage the use of the score based on the availability of patient data. In some models, no laboratory results were required. This is because some investigations are not covered by the Thai Universal Healthcare Coverage. Thus, it is possible that some patients do not have all lipid profile values [29,30]. The TCVR is widely used in Thailand and is incorporated into Thai national treatment guidelines [31]. The continuous variables, including age, SBP, WC, height, TC, HDL, and LDL, are entered in each model as continuous data. The variables used in the six models in the TCVR score are as follows:(1)Age, gender, DM, smoking status, SBP, and WC;(2)Age, gender, DM, smoking status, SBP, WC, and height;(3)Age, gender, DM, smoking status, SBP, and TC;(4)Age, gender, DM, smoking status, SBP, TC, and HDL;(5)Age, gender, DM, smoking status, SBP, HDL, and LDL;(6)Age, gender, DM, smoking status, SBP, and LDL.

### 2.4. Statistical Analysis

Descriptive analysis was used to describe participants’ characteristics. The association between patients’ demographic and MCI status was analyzed using Chi’s square and T-test. We created six prediction models using the variables presented in the six TCVR score models. A multivariable analysis was used. Each prediction model was evaluated using the area under the receiver operator characteristic curve (AuROC) to explore their clinical usefulness. The Hosmer–Lemeshow goodness of fit test was used for the determination of model calibration. For sensitivity analysis, every TCVR model was then applied to a subgroup of the population stratified by age (<65 or ≥65) and gender, as evidence has suggested that discriminative performance for cognitive performance may be modified by age and gender [32]. The subgroups included Male ≥ 65, Female ≥ 65, Male < 65, and Female < 65. The model at the lower end of the confidence interval for the AuROC of more than 0.5 would be considered to have discriminative properties in the detection of MCI [33]. STATA version 16 was used for analysis. For all statistical tests, a *p*-value of <0.05 was considered statistically significant.

### 2.5. Missing Data

The missing data are presented in Table 1. From the data that were needed for the models, the values missing were in the range of 1.37–4.11%. With the small amount of missing data, we assumed missing data occurred at random. We used a complete case analysis.

## 3. Results

### 3.1. Characteristics of the Study Participants

Table 1 demonstrates the participants’ characteristics. From a total of 421 eligible patients, there were 348 MCI and 73 non-MCI. The majority of patients were female (63.66%) and obese (55.58%). Hypertension status and MoCA scores were statistically significant between MCI and non-MCI groups (262 (75.29%) vs. 45 (61.64%), *p* = 0.017 and 19.54 ± 3.44 vs. 26.21 ± 1.21, *p* < 0.001, respectively). There were no significant differences in other characteristics and laboratory results across the categories of MCI status. There were only eight smokers in the sample, all with MCI; thus, this variable could not be used for model development.

### 3.2. Six TMCIR Models

Six models using different TCVR variables were developed. The adjusted odds ratio of each model is presented in Table 2. All six TCVR models had similar discriminative performance (AuROC between 0.58 and 0.61). Model 4, which consists of age, male gender, DM, SBP, TC, and HDL, provided the highest AuROC 0.61 (95% CI 0.53 to 0.68). Among the TCVR estimation models that exclude laboratory results (anthropometric-based models), Model 2 showed the best discriminative performance in distinguishing between an individual with MCI and non-MCI (AuROC 0.60, 95% CI 0.53 to 0.66). The model calibration was assessed using the Hosmer–Lemeshow goodness of fit test. The model fit statistics suggested that all the models fit the data; Model 1 X2 = 9.72, df = 8, *p* = 0.286; Model 2 X2 = 13.55, df = 8, *p* = 0.094; Model 3 X2 = 5.84, df = 8, *p* = 0.665; Model 4 X2 = 4.54, df = 8, *p* = 0.805; Model 5 X2 = 2.17, df = 8, *p* = 0.975; and Model 6 X2 = 6.73, df = 8, *p* = 0.566.

### 3.3. Sensitivity Analysis

In the sensitivity analysis stratified by age and gender, the discriminative performance of models that required only anthropometric data is shown in Table 3, and models that required some laboratory data are shown in Table 4. Among the anthropometric-based models, Model 2, consisting of age, gender, SBP, DM, WC, and height, showed the best risk prediction performance in the older age group (AuROC 0.69, 95% CI 0.58 to 0.79). Among the laboratory-based models, Model 4, consisting of age, gender, SBP, DM, TC, and HDL, provided the highest discriminative performance for the younger age group (AuROC 0.60, 95% CI 0.51 to 0.70).

## 4. Discussion

In our sample, the prevalence of MCI among patients with metabolic risks was as high as 82.7 percent. The discriminative performance (AuROC) of six different TCVR models utilizing commonly measured variables for CV risk assessment varied from 0.58 to 0.61. In the subgroup analysis, the anthropometric-based models demonstrated better discriminative performance in the older age group (≥65), whereas laboratory-based models provided higher discriminative performance for the younger age group (<65). However, the discriminative ability of the six models is still relatively poor and further model development and validation studies are needed to help better assess the risk of developing MCI for people with metabolic risks.

Detecting early stages of cognitive impairment such as MCI or early dementia might help physicians to provide better care for people with metabolic risks. The high prevalence of MCI in our sample is similar to the results of recent studies showing that the prevalence of MCI in individuals with metabolic risk factors was approximately between 65 and 75 percent [18,34]. This study has demonstrated that patients with metabolic risk factors have an increased risk of MCI and the TCVR models derived from commonly assessed cardiovascular risks could potentially be used to help predict the risk of early cognitive deficit, although with relatively poor discriminative performance. However, the potential to use CV risk assessment to help predict cognitive decline is supported by the literature [32]. High cardiovascular risk estimated by the Framingham Risk Score was shown to be related to worsening cognitive function in adults and the elderly. Early detection and treatment of mild cognitive impairment would help prevent further brain pathologies and slow the development of dementia [35]. The TCVR model could be used as a screening tool for MCI in patients with metabolic risk. After a positive screening for MCI, the patient needs to be further evaluated to establish a clear MCI diagnosis. An assessment, including the MoCA test as well as interviews with the patients and their families, should be performed [36]. Even though currently there is no specific pharmacological treatment for MCI, nonpharmacological interventions could potentially be useful. These interventions include reducing health risk behaviors (unhealthy diet, alcohol, sedentary lifestyle, stress, sleep deprivation), optimizing cardiovascular risk reduction, initiating mind-body exercise, discontinuing medications that might induce cognitive impairment, and advocating social engagement [37,38,39,40]. Moreover, early detection would provide the clinician with more time to discuss a long-term care plan with patients and their families.

The relatively poor discriminative property may be due to the limitation in identifying the exact causes of MCI. The underlying mechanism could be more than vascular etiology which is commonly associated with metabolic disease [41]. Non-vascular etiologies, for example, Alzheimer’s disease, Lewy body disease, and argyrophilic grains [42,43,44], may be the underlying cause; thus, the discriminative property using variables from cardiovascular risk assessment alone may not be so high. Additionally, the routine variables derived from cardiovascular risk assessment may not be sufficient. Other blood biomarkers that could be elevated in the preclinical stage of dementia, such as insulin [45], fibroblast growth factor 21 [26], lipocalin-2 [46], or trimethylamine N-oxide [47] may need to be considered to help increase the discriminative properties.

Interestingly, the algorithm without laboratory parameters was able to better predict the probability of MCI in older age individuals, especially in men. An independent risk factor for cognitive impairment in elderly people is anthropometric measurement. Obesity indicators, including body mass index and WC, are linked to an increased risk of cognitive decline. One study showed that the body mass index cut-off value of 26 kg/m^2^ or over for both genders, the WC cut-off value of 90 cm or over for men, and 82 cm or over for women were indicators for screening older adults who were at risk for MCI [48]. Furthermore, increased visceral fat is associated with insulin resistance which could affect brain function [49].

The model that included laboratory results, TC and HDL, on the other hand, appeared to be more beneficial in younger individuals. Several cardiovascular risk factors, including smoking, diabetic status, hypertension, and serum cholesterol in middle age, have previously been linked to late-life dementia [50,51]. In an elderly population, various factors may have accumulated over time that were not detectable at the time they aged. Older people showed a reduction in total LDL, and HDL cholesterol [52]. The aging process has an impact on cholesterol homeostasis. The decline in activity of acetyl CoA acetyltransferase 2, an essential enzyme in cholesterol metabolism, with an increase in hepatic free cholesterol is associated with decreased LDL in older age [53]. Decreased cholesterol levels may be affected by an age-related change in intestinal cholesterol absorption [54]. Additionally, individuals with frailty and sarcopenia have been shown to have an increased risk of dementia [55]. Lower serum cholesterol levels were also found in elderly patients with this condition [56]. Frailty may be related to the pathologic factors that lead to dementia in older adults, such as oxidative stress and inflammation. [57]. Another factor may be physiologic changes in association with the aging process with age-related lifestyle changes [58,59]. These findings agreed with the current study in that the model with laboratory testing showed a lower AuROC value when it was used to predict cognitive status in the older group.

This study demonstrated that there is the potential to develop an MCI risk assessment tool among patients with metabolic syndrome using the measurements obtained from routine cardiovascular risk assessments. Our results are in agreement with previous studies. The cardiovascular risk profile was linked to a decline in cognitive function and could be seen at as early as 35 years of age [32]. Another study found that Systematic Coronary Risk Evaluation was associated with cognitive decline from the neuropsychological tests in individuals with arterial hypertension [60]. The use of cardiovascular risk screening could potentially provide the ability to detect MCI. The strength of our study is that while most previous studies were conducted among older adults, our study provides information regarding MCI in adults of younger ages. Nonetheless, this study is not without its limitations. First, the diagnosis of MCI was based solely on the questionnaire. However, this method has been used widely in other relevant literature for the detection of MCI [61,62,63]. Second, some data from laboratory testing were not collected on the same visit as that of the questionnaire and anthropometric data assessment. However, we restricted the period of laboratory results to the six months prior to the questionnaire and anthropometric evaluation. This is to ensure that the laboratory results accurately reflect the patients’ status and are comparable to real-world practice, in which all laboratory results may not be completed in the same visit. Third, few people were still actively smoking, all with MCI. As a result, this variable was not included in any of the models. Finally, the relatively high prevalence of MCI in this population can affect the predictive values of our models, increasing the positive predictive value and decreasing the negative predictive value. However, the change in prevalence did not significantly affect the AuROC [64].

## 5. Conclusions

Patients with metabolic risk factors are at a higher risk for both major cardiovascular events and cognitive impairment. Early diagnosis of MCI could lead to the development of a treatment for prevention and delay the progression of the disease. This study suggests that the values and results obtained from routine cardiovascular risk assessment could potentially be used to develop a screening tool for MCI. However, as the discriminative ability of the six models is still relatively poor and may only be acceptable by using different models for different subgroups, further model development, and validation studies are needed.

## Figures and Tables

**Table 1 healthcare-10-01959-t001:** Participant characteristics.

Characteristic N = 421	Missing Values *n* (%)	MCI (*n* = 348)	Missing Values *n* (%)	Non-MCI (*n* = 73)	*p*-Value
Age (year, mean ± SD)	0	63.39 ± 7.15	0	62.00 ± 5.65	0.118
<65-year-old, *n* (%)		189 (54.31)		48 (65.75)	0.073
≥65-year-old, *n* (%)		159 (45.69)		25 (34.25)	
Male, *n* (%)	0	130 (37.36)	0	23 (31.51)	0.345
Female, *n* (%)	0	218 (62.64)	0	50 (68.49)	
Body mass index (kg/m^2^, mean ± SD)	0	26.17 ± 4.28	0	26.62 ± 5.43	0.435
Waist circumference (cm, mean ± SD)	0	89.09 ± 12.26	0	88.63 ± 13.75	0.777
SBP (mmHg, mean ± SD)	0	135.56 ± 15.45	0	134.26 ± 15.17	0.512
DBP (mmHg, mean ± SD)	0	76.65 ± 9.59	0	77.59 ± 9.66	0.449
Underlying disease					
Hypertension, *n* (%)	0	262 (75.29)	0	45 (61.64)	0.017
Dyslipidemia, *n* (%)	0	259 (74.43)	0	54 (73.97)	0.936
Type 2 Diabetes, *n* (%)	0	151 (43.39)	0	25 (34.25)	0.150
Alcohol drinker, *n* (%)	0	81 (24.14)	0	12 (16.44)	0.154
Smoking, *n* (%)	0	8 (2.30)	0	0 (0)	0.191
Assessment					
MoCA score (mean ± SD)	0	19.54 ± 3.44	0	26.21 ± 1.21	<0.001
Laboratory results					
FBS (mg/dL, mean ± SD)	49 (14.08)	119.92 + 45.52	65 (10.96)	109.30 + 30.79	0.074
TG (mg/dL, mean ± SD)	6 (1.72)	128.65 ± 71.62	1 (1.37)	117.71 ± 57.22	0.224
TC (mg/dL, mean ± SD)	7 (2.01)	171.56 ± 38.24	3 (4.11)	170.66 ± 38.75	0.857
HDL (mg/dL, mean ± SD)	6 (1.72)	55.36 ± 15.44	1 (1.37)	58.93 ± 15.57	0.075
LDL (mg/dL, mean ± SD)	6 (1.72)	105.25 ± 35.52	2 (2.74)	102.92 ± 33.11	0.610

Data are presented as absolute numbers and percentages or mean and standard deviation. Abbreviations: DBP, diastolic blood pressure; FBS, fasting blood sugar; HDL, high-density lipoprotein cholesterol; LDL, low-density lipoprotein cholesterol; MCI, mild cognitive impairment; MoCA, Montreal Cognitive Assessment tool; SBP, systolic blood pressure; TC, total cholesterol; TG, triglyceride; WC, waist circumference.

**Table 2 healthcare-10-01959-t002:** Prediction models for MCI regarding variables from TCVR.

Model	aOR (95% CI)					
	M1 (*n* = 421)	M2 (*n* = 421)	M3 (*n* = 411)	M4 (*n* = 409)	M5 (*n* = 412)	M6 (*n* = 413)
Age (year)	1.02 (0.99–1.06)	1.02 (0.98–1.06)	1.03 (0.99–1.07)	1.03 (0.98–1.07)	1.03 (0.99–1.07)	1.03 (0.99–1.07)
Male	1.29 (0.74–2.24)	1.77 (0.83–3.77)	1.25 (0.71–2.19)	1.20 (0.67–2.14)	1.25 (0.70–2.21)	1.35 (0.77–2.36)
DM	1.45 (0.82–2.56)	1.40 (0.79–2.48)	1.57 (0.88–2.80)	1.47 (0.82–2.65)	1.43 (0.80–2.55)	1.55 (0.88–2.73)
SBP (mmHg)	1.00 (0.99–1.02)	1.00 (0.99–1.02)	1.00 (0.99–1.02)	1.00 (0.99–1.02)	1.00 (0.99–1.02)	1.00 (0.99–1.02)
WC (cm)	0.99 (0.97–1.02)	1.00 (0.97–1.02)				
Height (cm)		0.97 (0.92–1.02)				
TC (mg/dL)			1.00 (1.00–1.01)	1.01 (1.00–1.01)		
HDL (mg/dL)				0.99 (0.97–1.00)	0.99 (0.97–1.01)	
LDL (mg/dL)					1.01 (1.00–1.01)	1.01 (1.00–1.01)
ROC	0.58 (0.51–0.65)	0.60 (0.53–0.66)	0.59 (0.52–0.67)	0.61 (0.53–0.68)	0.60 (0.53–0.67)	0.59 (0.51–0.66)

Models’ Components: M1: Age, gender, DM, SBP, and WC. M2: Age, gender, DM, SBP, WC, and height. M3: Age, gender, DM, SBP, and TC. M4: Age, gender, DM, SBP, TC, and HDL. M5: Age, gender, DM, SBP, HDL, and LDL. M6: Age, gender, DM, SBP, and LDL. Smoking status was omitted from the models due to the small number of current smokers. Abbreviations: aOR: adjusted odds ratio; CI: confidence interval; DM: diabetes mellitus; HDL: high-density lipoprotein cholesterol; LDL: low-density lipoprotein cholesterol; ROC: receiver operating characteristic; SBP: systolic blood pressure; TC: total cholesterol; WC: waist circumference.

**Table 3 healthcare-10-01959-t003:** Subgroup analysis of discriminative performance of anthropometric-based models.

Model	Subgroup Analysis	*n*	ROC Area	95% CI	
Model 1: Age, gender, DM, SBP and WC	All ≥ 65	184	0.63	0.52	0.73
	Male ≥ 65	74	0.67	0.49	0.85
	Female ≥ 65	110	0.57	0.42	0.71
	All < 65	237	0.52	0.43	0.61
	Male < 65	79	0.46	0.29	0.64
	Female < 65	158	0.53	0.42	0.64
Model 2: Age, gender, DM, SBP WC, and height	All ≥ 65	184	0.69	0.58	0.79
	Male ≥ 65	74	0.81	0.65	0.97
	Female ≥ 65	110	0.59	0.46	0.73
	All < 65	237	0.52	0.43	0.62
	Male < 65	79	0.46	0.30	0.63
	Female < 65	158	0.54	0.44	0.65

Smoking status was omitted from the models due to the small number of current smokers. Abbreviations: aOR: adjusted odds ratio; CI: confidence interval; DM: diabetes mellitus; ROC: receiver operating characteristic; SBP: systolic blood pressure; WC: waist circumference.

**Table 4 healthcare-10-01959-t004:** Subgroup analysis of discriminative performance of laboratory-based models.

Model	Subgroup Analysis	*n*	ROC Area	95% CI	
Model 3: Age, gender, DM, SBP and TC	All ≥ 65	181	0.59	0.47	0.71
	Male ≥ 65	73	0.53	0.37	0.70
	Female ≥ 65	108	0.61	0.45	0.77
	All < 65	230	0.56	0.47	0.66
	Male < 65	78	0.56	0.40	0.72
	Female < 65	152	0.57	0.45	0.69
Model 4: Age, gender, DM, SBP TC, and HDL	All ≥ 65	180	0.57	0.44	0.70
	Male ≥ 65	72	0.48	0.30	0.66
	Female ≥ 65	108	0.61	0.45	0.76
	All < 65	229	0.60	0.51	0.70
	Male < 65	77	0.64	0.48	0.81
	Female < 65	152	0.59	0.47	0.70
Model 5: Age, gender, DM, SBP HDL and LDL	All ≥ 65	180	0.57	0.44	0.70
	Male ≥ 65	71	0.50	0.32	0.69
	Female ≥ 65	109	0.59	0.43	0.76
	All < 65	232	0.59	0.50	0.68
	Male < 65	78	0.65	0.48	0.82
	Female < 65	154	0.55	0.44	0.66
Model 6: Age, gender, DM, SBP, and LDL	All ≥ 65	180	0.56	0.43	0.69
	Male ≥ 65	71	0.52	0.34	0.69
	Female ≥ 65	109	0.59	0.42	0.77
	All < 65	233	0.56	0.47	0.65
	Male < 65	78	0.59	0.42	0.75
	Female < 65	155	0.55	0.44	0.67

Smoking status was omitted from the models due to the small number of current smokers. Abbreviations: aOR: adjusted odds ratio; CI: confidence interval; DM: diabetes mellitus; HDL: high-density lipoprotein cholesterol; LDL: low-density lipoprotein cholesterol; ROC: receiver operating characteristic; SBP: systolic blood pressure; TC: total cholesterol; WC: waist circumference.

## Data Availability

The data underlying this article will be shared upon reasonable request to the corresponding author.

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
