# Peer review of "Risk Prediction Performance of the Thai Cardiovascular Risk Score for Mild Cognitive Impairment in Adults with Metabolic Risk Factors in Thailand"

_healthcare, 2022, doi:10.3390/healthcare10101959_

Round 1

Reviewer 1 Report (Previous Reviewer 1)

The authors have clearly responded to my comments and suggestions. 

Author Response

Thank you.

Best regards,

Kanokporn Pinyopornpanish

Reviewer 2 Report (New Reviewer)

Thank you for submitting the manuscript for review.  The publication shows an alternative model for risk assessment of MCI. It shows an interesting approach using routine data collected during a cardiovascular check. The discriminative ability of the six models is still relatively poor and further model development and validation studies are needed. However, this only makes sense if MCI can be counteracted therapeutically at an early stage. Here, the discussion should be intensified to support the importance of your tool. While reading the paper, few thoughts occurred to me, which I would now like to present to you. Best success with the further progress.

 Introduction

1)     Line 54. Please add a definition of MCI. Which characteristics are typical?

2)     Lines 69 amd 71: Are there double spaces before "For" and "This"?

 Dataset and Participants

3)     Lines 81/82: The ethics committee numbers differ significantly. Is that a number for your data collection or also for the study according to reference 25? This is written somewhat unclearly.

4)     Please explain the recruitment of the subjects? From which medical facilities were recruited? e.g., neurological or geriatric facilities? Were subjects aware that they had MCI?

 Statistical Analysis

5)     Explain the reason for the classification into the age groups.

 Results

6)     Table 1: Please delete the points after kg (BMI) and cm (WC).

 Discussion

7)     Lines 200/201: See comment #5) Could the high prevalences of MCI result of selection bias?

8)     2nd Paragraph: What is the therapy against MCI? (Apart from optimizing cardiovascular risk factors. I imagine it will be difficult in older age.) Could the course of MCI be improved if screening for MCI is positive? I recommend discussing this because it increases the importance of your tool.

 References

9)     Please adapt reference no. 21 and 28 to the guidelines (jounal names), e. g. Write only initial letters in capital letters or same font size.

10) Please adjust the bibliography: abbreviated journal names according to author guidelines.

Author Response

Responses to Reviewer’s Comments

Point 1: Thank you for submitting the manuscript for review. The publication shows an alternative model for risk assessment of MCI. It shows an interesting approach using routine data collected during a cardiovascular check. The discriminative ability of the six models is still relatively poor and further model development and validation studies are needed. However, this only makes sense if MCI can be counteracted therapeutically at an early stage. Here, the discussion should be intensified to support the importance of your tool. While reading the paper, few thoughts occurred to me, which I would now like to present to you. Best success with the further progress.

Response 1: We would like to thank the reviewer for the encouragement and comments. The manuscript has been revised according to the reviewers’ suggestions and we hope that our response addresses the reviewers’ comments. We have also sent our manuscript for language editing. Below are our point-by-point responses to the reviewer’s comments.

Point 2: Line 54. Please add a definition of MCI. Which characteristics are typical?

Response 2: Thank you for your suggestion. We have added the sentence. It now reads “MCI is the condition that lies between normal cognitive aging and dementia. It is defined as evidence of progressive cognitive decline combined with preservation of functional independence [17].” (Page 2 Line 54-57).

Point 3: Lines 69 and 71: Are there double spaces before "For" and "This"?

Response 3: We apologize for our mistake. We have corrected the space as suggested.  

Point 4: Lines 81/82: The ethics committee numbers differ significantly. Is that a number for your data collection or also for the study according to reference 25? This is written somewhat unclearly.

Response 4: We would like to apologize for this unclear statement. The ethics committee numbers for a previous study examining the possible link between fibroblast growth factor 21 and cognitive decline among patients with metabolic risk factors (reference 26, was 25) and additional collected data to increase the sample size and power between 1 May 2020 to 31 May 2021 were 2671 and 08246, respectively. We have rephrased the sentence. It now reads “The study protocols were approved by the Research Ethics Committee of the Faculty of Medicine, Chiang Mai University, for the previous study (No. 02671) and additional data collection (No. 08246).” (Page 2 Line 84-86).

Point 5: Please explain the recruitment of the subjects? From which medical facilities were recruited? e.g., neurological or geriatric facilities? Were subjects aware that they had MCI?

Response 5: Thank you for pointing this out. We recruited the subjects from the outpatient clinics in our hospital which provide care for patients with metabolic risks. These clinics included family medicine clinic, general medicine clinic, endocrinology clinic, and cardiology clinic. Subjects were not aware that they had MCI prior to the enrollment period. It now reads “The subjects were recruited from the outpatient clinics at Maharaj Nakorn Chiang Mai Hospital which provide care for patients with metabolic risks. These clinics included the family medicine clinic, general medicine clinic, endocrinology clinic, and cardiology clinic. Subjects had not been diagnosed with MCI prior to the enrollment period.” (Page 2 Line 86-90).

Point 6: Explain the reason for the classification into the age groups.

Response 6: Literature suggests that the prediction performance may be affected by age. Therefore, we classified them as adults (<65) and old adults (65 and older). We have included the statement in our statistical analysis subsection. The statement reads “For sensitivity analysis, every TCVR model was then applied to a subgroup of the population stratified by age (< 65 or ≥ 65) and gender, as evidence has suggested that discriminative performance for cognitive performance may be modified by age and gender [32].” (Page 3 Line 137-140).

Point 7: Table 1: Please delete the points after kg (BMI) and cm (WC).

Response 7: We apologize for our mistake. We deleted the points after kg (BMI) and cm (WC).

Point 8: Lines 200/201: See comment #5) Could the high prevalences of MCI result of selection bias?

Response 8: Thank you for the comment. As stated in response to comment 5, we recruited the subjects from the clinics that provide care for patients with metabolic risks, not the neurology clinic. However, this population is more likely to have a higher prevalence compared to the general population. This issue has been discussed in our discussion section. The statement reads “The high prevalence of MCI in our sample is similar to recent studies showing that the prevalence of MCI in individuals with metabolic risk factors was approximately between 65 to 75 percent [18,34]. This study has demonstrated that patient with metabolic risk factors have an increased risk of MCI and the TCVR models derived from commonly assessed cardiovascular risks could potentially be used to help predict the risk of early cognitive deficit but still with relatively poor discriminative performance. However, the potential to use CV risk assessment to help predict cognitive decline is supported by the literature [32]. (Page 7 Line 219-227)

Point 9: 2nd Paragraph: What is the therapy against MCI? (Apart from optimizing cardiovascular risk factors. I imagine it will be difficult in older age.) Could the course of MCI be improved if screening for MCI is positive? I recommend discussing this because it increases the importance of your tool.

Response 9: Thank you for your insightful comment. Currently, there is no specific treatment for MCI. However, the growing evidence suggests the potential benefit of early detection and risk reduction at the MCI stage. Apart from optimizing cardiovascular risk factors, there are some further useful interventions that we have added to our discussion. The statement reads “Early detection and treatment of mild cognitive impairment would help prevent further brain pathologies and slow the development of dementia [35]. The TCVR model could be used as a screening tool for MCI in patients with metabolic risk. After a positive screening for MCI, the patient needs to be further evaluated to establish a clear MCI diagnosis. The assessment, including the MoCA test as well as interviews with the patients and their families, should be done [36]. Even though currently there is no specific pharmacological treatment for MCI, nonpharmacological interventions could potentially be useful. These interventions include reducing health risk behaviors (unhealthy diet, alcohol, sedentary lifestyle, stress, sleep deprivation), optimizing cardiovascular risk reduction, initiating mind-body exercise, discontinuing medications that might induce cognitive impairment, and advocating social engagement [37-40]. Moreover, early detection would provide the clinician with more time to discuss a long-term care plan with patients and their families.” (Page 7 Line 228-240).

Point 10: Please adapt reference no. 21 and 28 to the guidelines (jounal names), e. g. Write only initial letters in capital letters or same font size.

Response 10: We have adjusted the bibliography no. 22 (was 21) and 29 (was 28).

22. Ridtidat, R.; Komanasin, N.; Mogkolwongroj, P. Assessment of cardiovascular disease risk by the Rama-EGAT heart score in staff of Songkhla hospital. Journal of Medical Technology and Physical Therapy 2015, 27, 14-27.

29. Thai Atherosclerosis Society. Clinical Practice Guideline on Pharmacologic Therapy of Dyslipidemia for Atherosclerotic Cardiovascular Disease Prevention, The Royal College of Physicians of Thailand (RCPT), 2016.

Point 11: Please adjust the bibliography: abbreviated journal names according to author guidelines.

Response 11: Thank you for your suggestion. We have adjusted the bibliography according to author guidelines.

This manuscript is a resubmission of an earlier submission. The following is a list of the peer review reports and author responses from that submission.

Round 1

Reviewer 1 Report

11. Table 1, Table 2, Table 4 Model 6 (ALL<65) There are no Female data. Please clarify.

22. Authors may need to compare results with other literature in the discussion section.

  3. Line 227;  26 kg/m2  “2” should be superscript.

Reviewer 2 Report

Abstract: The abstract is well structured, and it contains the main information of the study.

Introduction: The introduction identifies the problem that is being addressed in the manuscript. I can clearly understand the purpose of the study. Only exception is the sentence at line 58-59, can you rephrase or explain a little better.

Methods: This section contains enough information to understand and possibly repeat the study. Authors can you provide some more explanation or clarify the following points:

a)   Statistical analysis:

a.       Consider also to compare the roc area results; its is necessary to have all those models, are they informative, in my opinion this is not true (see comments in the results sections).

b.       could you describe more accurately the approach to table 3 and 4

Results: this is the section where I have more problems:

a)       Line 136-138 could you describe a little better

b)      Table 2, Probably you have TC, HDL, LDL as predictors, all the other variables considered were not associate with MCI.

c)       Looking at ROC results the most efficient model is the M1, less variables considered with a quite good ROC area, and probably no differences could be found in the comparison with others ROC models.

d)      Table 3 and 4 are not so clear what authors would like to demonstrate. Again, are those variables significantly associated with MCI.

Discussion

Line 207-209; In my opinion what you could demonstrate is only that, patients with MS, have an increased risk of MCI, and probably could develop a form of dementia (ischemic more probably), but this is not knew.

Moreover, as you correctly stated you have a very low discriminatory performance, therefore the plus of your paper is at least questionable.

Reviewer 3 Report

Please find my comments in the file attached.
